# RNAseq Analysis of FABP4 Knockout Mouse Hippocampal Transcriptome Suggests a Role for WNT/β-Catenin in Preventing Obesity-Induced Cognitive Impairment

**DOI:** 10.3390/ijms24043381

**Published:** 2023-02-08

**Authors:** Simon W. So, Joshua P. Nixon, David A. Bernlohr, Tammy A. Butterick

**Affiliations:** 1Minneapolis Veterans Affairs Health Care System, Minneapolis, MN 55417, USA; 2Department of Neuroscience, University of Minnesota Twin Cities, Minneapolis, MN 55455, USA; 3Department of Food Science and Nutrition, University of Minnesota Twin Cities, St. Paul, MN 55108, USA; 4Department of Surgery, University of Minnesota Medical School, Minneapolis, MN 55455, USA; 5Department of Biochemistry, Molecular Biology, and Biophysics, University of Minnesota Twin Cities, Minneapolis, MN 55455, USA

**Keywords:** obesity, transcriptome, hippocampus, long-term potentiation, cognitive decline, inflammation, microglia, WNT/β-Catenin

## Abstract

Microglial fatty-acid binding protein 4 (FABP4) is a regulator of neuroinflammation. We hypothesized that the link between lipid metabolism and inflammation indicates a role for FABP4 in regulating high fat diet (HFD)-induced cognitive decline. We have previously shown that obese FABP4 knockout mice exhibit decreased neuroinflammation and cognitive decline. FABP4 knockout and wild type mice were fed 60% HFD for 12 weeks starting at 15 weeks old. Hippocampal tissue was dissected and RNA-seq was performed to measure differentially expressed transcripts. Reactome molecular pathway analysis was utilized to examine differentially expressed pathways. Results showed that HFD-fed FABP4 knockout mice have a hippocampal transcriptome consistent with neuroprotection, including associations with decreased proinflammatory signaling, ER stress, apoptosis, and cognitive decline. This is accompanied by an increase in transcripts upregulating neurogenesis, synaptic plasticity, long-term potentiation, and spatial working memory. Pathway analysis revealed that mice lacking FABP4 had changes in metabolic function that support reduction in oxidative stress and inflammation, and improved energy homeostasis and cognitive function. Analysis suggested a role for WNT/β-Catenin signaling in the protection against insulin resistance, alleviating neuroinflammation and cognitive decline. Collectively, our work shows that FABP4 represents a potential target in alleviating HFD-induced neuroinflammation and cognitive decline and suggests a role for WNT/β-Catenin in this protection.

## 1. Introduction

Midlife obesity is associated with earlier onset of Alzheimer’s disease (AD) and increased AD neuropathology [1,2,3]. Obesity is often characterized by chronic low-level inflammation, along with metabolic disorders such as type-2 diabetes mellitus and insulin resistance [4,5,6]. Insulin resistance is accompanied by increased fatty acid transport and circulation. Fatty acid binding protein 4 (FABP4; also known as adipocyte protein-2; aP2) is a fatty acid binding protein with roles in fatty acid transport and lipid metabolism. FABP4 knockout in adipocytes is associated with reduced inflammation, insulin resistance, and metabolic dysfunction [7]. Overconsumption of a high fat diet (HFD), specifically those rich in saturated fatty acids (SFA), has been shown to exacerbate neuroinflammation, neurodegeneration, and cognitive impairment [4,6,8,9,10,11,12,13,14,15]. Studies indicate that this is influenced more by dietary SFA content than by total calories consumed [4,15].

The FABP4 knockout mouse model (AKO) has allowed an increasingly comprehensive understanding of lipid metabolism, diabetes, and peripheral immune response [16,17,18,19,20]. Unlike obese wild type controls, AKO mice present with decreased TNF-α (a biomarker of obesity-related insulin resistance) in adipose tissue [5]. Fatty acid metabolism plays a central role in immunometabolic diseases, including diet-induced obesity and metabolic syndrome [16,21,22]. We hypothesized that the link between lipid metabolism and inflammation indicates a role for FABP4 in regulating HFD-induced cognitive decline. We have previously shown that AKO mice are protected against HFD-induced hippocampal proinflammatory cytokine expression and memory deficits [23,24]. These mice were fed HFD for 12 weeks starting at 15 weeks of age. Herein, we investigated the transcriptomics of the AKO model fed HFD compared to wild type (WT) mice fed the same diet. As our previous research showed no differences in hippocampal inflammation and cognitive decline between groups fed normal chow diet (NC), nor between HFD-fed and NC-fed AKO groups [23], only HFD hippocampal transcriptomes were analyzed. Our present results show that AKO mice present with a neuroprotective hippocampal transcriptome that correlates with a decrease in proinflammatory signaling, ER stress, apoptosis, and cognitive decline, and an increase in neurogenesis, synaptic plasticity, long-term potentiation, and spatial working memory. This is accompanied by an increase in WNT/β-Catenin signaling, suggesting a role for the pathway in preventing HFD-induced neuroinflammation and cognitive decline.

## 2. Results

### 2.1. Transcript Analysis

We determined the transcripts that were differentially expressed in HFD-fed AKO mice compared to HFD-fed WT mice. AKO and WT mice were fed 60% HFD for 12 weeks starting at 15 weeks of age. Hippocampal tissue was dissected and RNAseq was performed to measure differentially expressed transcripts. Transcript analysis showed 1793 differentially expressed transcripts at log10 adjusted *p*-value (*p*adj) ≤ 0.05 (Figure 1).

### 2.2. Downregulated Transcripts in HFD-Fed AKO Compared to HFD-Fed WT Mice

We determined the transcripts that were most downregulated in HFD-fed AKO mice compared to HFD-fed WT mice (LFC ≤ −1.00, *p*adj ≤ 0.001). There were 12 transcripts in total (Table 1). HFD-fed AKO mice have downregulated transcripts associated with negative regulation of Wnt/β-Catenin signaling [25] and neurite outgrowth [26] along with positive regulation of cognitive decline [27], proinflammatory cytokine signaling [28,29], amyloid beta plaque formation [30], and ER stress [31]. Downregulated transcripts also have roles associated with lipid metabolism [32] and GABA signaling [33,34].

### 2.3. Upregulated Transcripts in HFD-Fed AKO Compared to HFD-Fed WT Mice

We determined the transcripts that were most upregulated in HFD-fed AKO mice compared to HFD-fed WT mice (LFC ≥ 1.00, *p*adj ≤ 0.001). There were 19 transcripts in total (Table 2). AKO mice fed HFD have upregulated transcripts associated with negative regulation of apoptosis [42,43,44], neuroinflammation [45], and ER stress [42,43] along with positive regulation of long-term potentiation [46,47,48], spatial working memory [49], synaptic plasticity [50,51], neurogenesis [52,53], and Wnt/β-Catenin signaling [54].

### 2.4. Differentially Expressed Pathways in HFD-Fed AKO Compared to HFD-Fed WT Mice

Using Reactome molecular pathway analysis, we determined the differentially expressed pathways in HFD-fed AKO mice compared to HFD-fed WT mice (FDR ≤ 0.01). There were 39 pathways in total. Nine of these pathways were unrelated to ribosomal protein subunits and were examined further for transcript function (Appendix A). We further condensed these results into pathways that relate to metabolism and Wnt/β-Catenin signaling (Table 3). Transcripts associated with the electron transport chain [68,69,70,71,72,73,74,75] and citric acid cycle [76,77,78,79] were downregulated and upregulated, respectively. Transcripts relating to Wnt/β-Catenin signaling were upregulated [80,81,82,83,84,85].

### 2.5. Differentially Expressed Microglial Markers in HFD-Fed AKO Compared to HFD-Fed WT Mice

We determined the differentially expressed transcripts that have been identified as microglial markers (*p*adj ≤ 0.05). There were 10 transcripts in total (Table 4). AKO mice fed HFD have transcripts associated with microglial markers differentially expressed. Upregulated transcripts have roles in the regulation of M2 microglial transcriptome activation [87], ER/Golgi transport [88,89], dopaminergic neuronal survival [90], FcγR-dependent phagocytosis [91], and long-term potentiation [92,93].

### 2.6. β-Catenin Western Blot

We performed Western blotting for nuclear β-Catenin in HFD-fed AKO and WT mouse hippocampal tissue. Results showed that HFD-fed AKO mice had increased expression of nuclear β-Catenin compared to WT (Figure 2; *p* = 0.03).

## 3. Discussion

Obesity is known to be associated with peripheral and central chronic low-grade inflammation [99]. Peripherally, HFD is linked to low-grade systemic inflammation associated with the development of metabolic disorders such as type 2 diabetes mellitus [100]. In the brain, diets high in SFAs are known to be associated with neuroinflammation, cognitive impairment, and the development of neurodegenerative diseases [101]. An increase in circulating free fatty acids (FFAs) disrupts metabolic homeostasis [102]. This increase in FFAs can also activate Toll-like receptor 4 (TLR4) in microglia, leading to ER stress and increased inflammatory cytokine expression [103]. In turn, increased ER stress and neuroinflammation can lead to cognitive deficits and neurodegenerative diseases such as AD [102]. The HFD-fed FABP4 knockout model has revealed an altered hippocampal transcriptome that is shown to have changes in metabolic pathways and be neuroprotective against HFD-induced neuroinflammation and cognitive decline.

In the present study, the AKO model has shown an altered hippocampal transcriptome with changes in metabolic pathways. AKOs had a decrease in pathways relating to the electron transport chain (ETC). These pathways were “formation of ATP by chemiosmotic coupling”, “cristae formation”, “complex I biogenesis”, “respiratory electron transport, ATP synthesis by chemiosmotic coupling, and heat production by uncoupling proteins”, and “respiratory electron transport” (Appendix A). Importantly, the major downregulated transcripts in these pathways encode for subunits of NADH:ubiquinone oxidoreductase (Complex I) and ATP synthase (Complex V) of the ETC. Partial inhibition of Complex I has been shown to improve energy homeostasis, synaptic activity, long-term potentiation, and cognitive function while reducing oxidative stress and inflammation in the brain [68]. Targeting Complex V has been shown to decrease expression of an aged and dementia-associated transcriptome [86]. AKOs had an increase in the “citric acid (TCA) cycle and respiratory electron transport” pathway (Table 3). It is important to note that transcripts relating to the tricarboxylic acid cycle (TCA) were increased. These transcripts include *Aco2*, *Cs*, *Ogdh*, and *Sdha*. Downregulation of these transcripts have been observed in neurodegenerative transcriptomes and some have been associated with oxidative stress and insulin resistance [74,76,77,78]. Our previous work has shown that FABP4 knockout confers protection against ROS production and ER stress through an increase in UCP2 [20]. As such, our present results provide further evidence that FABP4 knockout leads to protective metabolic changes in HFD-fed mice. The AKO model has also shown an increase in a pathway relating to Golgi transport. This pathway is named “transport to the Golgi and subsequent modification” (Appendix A). The upregulated genes in this pathway include regulators of ER to Golgi transport [104,105] and a decrease in some of these transcripts have been associated with ER stress [106] and neurodegenerative diseases [107,108]. As Golgi function and transport is important for lipid metabolism [109], changes in transcripts relating to these pathways provide further insight on the neuroprotection conferred via FABP4 knockout. Differentially expressed genes relating to lipid metabolism in AKOs were *Apoa2* (Table 1) and *Sort1* (Table 2). *Apoa2* was downregulated in AKOs and encodes for a high-density lipoprotein. Certain polymorphisms of *Apoa2* are known to be associated with obesity [110]. *Sort1* was upregulated in AKOs and has been shown to modulate low-density lipoprotein uptake in macrophages [111]. Together, these changes in metabolic and Golgi transport pathways, along with genes associated with changes in lipid metabolism, show that HFD-fed AKO mice have a hippocampal transcriptome consistent with neuroprotection.

Previously, we have shown that HFD-fed AKO mice exhibit an alleviation of hippocampal inflammatory cytokine signaling compared to HFD-fed WT mice [23]. In the present study, HFD-fed AKO mice had transcriptome changes consistent with a decrease in neuroinflammation. AKOs had decreased expression of *Tpx2* and *Dmd* (Table 1), along with increased expression of *S1pr3* (Table 2). Silencing of *Tpx2* has been shown to decrease inflammation as marked by decreases in TNF-α, IL-6, and IL-8 protein levels [28]. Mice deficient of dystrophin, the protein encoded by *Dmd*, exhibit decreased infiltration of CD3+ T cells [29]. Knockdown of *S1pr3* has been shown to exacerbate neuroinflammation as measured by IBA1 and TNF-α [45]. HFD-fed AKO mice also had transcriptome changes consistent with a decrease in ER stress. AKOs had decreased expression of *Lonp2* (Table 1), and increased expression of *Hspa5* and *Smpd4* (Table 2). ER stress is known to cause a decrease in LONP2 expression [31]. HSPA5 is known to be a regulator of ER stress in the brain [42] and loss of SMPD4 has been shown to induce ER stress in the brain [43]. HFD is known to cause ER stress in the hippocampus [112] and ER stress can trigger inflammatory responses in the brain, leading to neurodegeneration [113]. Therefore, the transcriptomic changes relating to decreases in ER stress provide a potential mechanism for the decrease in neuroinflammation.

We have also previously shown that HFD-fed AKO mice exhibit an alleviation of cognitive decline and memory deficits compared to HFD-fed WT mice [23]. In the present study, HFD-fed AKO mice had transcriptome changes consistent with a decrease in cognitive decline. AKOs had decreased expression of *Apoa2* (Table 1). Increased APOA2 expression is associated with cognitive impairment and late-life dementia [27]. AKOs also had transcriptome changes consistent with a decrease in apoptosis. Expression of *Hspa5* and *Smpd4* were increased (Table 2). HSPA5 is known to be an antiapoptotic factor for cells undergoing ER stress [42]. Loss of SMPD4 has been shown to lead to apoptosis under stress conditions [43]. As apoptosis leading to neuronal cell death has been associated with AD pathogenesis [114], an increase in antiapoptotic factors could contribute to a neuroprotective profile against cognitive decline. AKOs also had a transcriptome negatively associated with AD. Expression of *Eif4e2* and *Wtap* were decreased (Table 1), and expression of *Atp2a2* was increased (Table 2). The phosphorylated state of EIF4E is associated with hyperphosphorylated tau and AD [36]. WTAP is a component of the complex responsible for m6A methylation [40], which is increased in AD mice [41]. Expression of ATP2A2 has been shown to be decreased in the AD brain [58]. As cognitive decline is a hallmark of AD, a transcriptome negatively associated with AD further reveals the neuroprotective profile of the AKO hippocampus.

The transcriptomic changes in HFD-fed AKO mice also reveals differentially expressed transcripts that relate to neurite outgrowth and neurogenesis. Expression of *Tmod2* was decreased (Table 1), and expression of *Gpr68* and *Sufu* were increased (Table 2). Knockdown of *Tmod2* leads to an increase in neurite outgrowth in vitro [26]. There is evidence to show that GPR68 has a positive role in neurogenesis [52]. Deletion of *Sufu* has been shown to decrease neurogenesis in the dentate gyrus [53]. As a decrease in neurogenesis is linked to cognitive deficits and neurodegenerative disease [115], transcriptomic changes that promote neurogenesis would further confer protection against cognitive decline. HFD-fed AKO mice also had transcriptomic changes relating to synaptic plasticity and long-term potentiation. Expression of *Brd1*, *Adra2a*, *Mdga1*, and *Grik2* were increased (Table 2). The BRD1 protein is needed for H3K14 acetylation [63], which is associated with synaptic plasticity [50]. The activation of α2-adrenergic receptors has been shown to promote long-term potentiation in the mouse brain [46]. Mice lacking MDGA1 and GRIK2 have exhibited compromised hippocampal long-term potentiation [47,48]. Long-term potentiation and synaptic plasticity are critical for forming memories and preventing cognitive impairment [116]. The AKO hippocampal transcriptome further reveals protection against cognitive decline by having upregulated transcripts that promote synaptic plasticity.

The WNT/β-Catenin signaling pathway has been shown to be neuroprotective against cognitive decline and neurodegenerative disease. Mechanisms that confer this protection include increasing neuronal survival, neurogenesis, synaptic plasticity, and blood–brain barrier integrity, as well as decreasing amyloid-β production and tau hyperphosphorylation [83,117,118,119]. HFD-fed AKO mice exhibited transcript and pathway changes consistent with an increase in WNT/β-Catenin signaling. After *Fabp4*, the next most downregulated transcript was *Sox6* (Table 1), which has been shown to inhibit WNT/β-Catenin signaling in adipocytes [25]. The third most upregulated transcript was *Nox1* (Table 2), which has been shown to increase WNT/β-Catenin signaling [54]. Reactome analysis revealed that the most upregulated pathway was “disassembly of the destruction complex and recruitment of AXIN to the membrane” (Table 3). Within this pathway, transcripts that were increased included *Ctnnb1*, *Dvl3*, *Fzd1*, *Ppp2cb*, *Ppp2r1a*, *Ppp2r5a*, and *Ppp2r5e* (Table 3). These transcripts encode for proteins that are positive regulators of WNT/β-Catenin signaling, and have roles in neuronal survival, synaptic plasticity, neuronal differentiation, and improved spatial memory [80,83,84,85]. *Ccnd1*, a target of WNT/β-Catenin signaling [120], was also found to be increased in AKOs (Log FC: 0.63, *p*adj: 3.62 × 10^−2^). As CCND1 has been shown to be a positive regulator of neurogenesis [121], the increase in this target gene further supports the neuroprotective profile of AKOs. To verify the increase in WNT/β-Catenin signaling, we performed Western blotting for nuclear β-Catenin in HFD-fed AKO and WT mouse hippocampal tissues. Results showed that AKO mice had an 80% increase in nuclear β-Catenin expression compared to WT (Figure 2). This is higher than the increase in *Ctnnb1* transcript, which was 30% (Table 3). These results further show that loss of FABP4 confers neuroprotection against HFD-induced cognitive decline and suggests a role for WNT/β-Catenin in this protection. It is important to note that *Csnk1a1*, *Csnk1g2*, and *Gsk3b* were also upregulated in this pathway (Table 3). These transcripts encode for proteins that are canonically negative regulators of WNT/β-Catenin signaling. These proteins are known to have roles in other pathways in the brain. Casein kinase 1 is known to be a negative regulator of TGF-β [82], whose overproduction in the brain has been linked with glucose intolerance [122]. GSK3B also has many roles in the brain including regulating neurogenesis, axon growth, and synaptic plasticity [123]. These roles may explain why these transcripts were upregulated in AKOs.

FABP4 is expressed in peripheral macrophages and microglia. In the periphery, knockout of FABP4 in macrophages leads to a decrease in inflammatory cytokine secretion [124]. We have previously shown that FABP4 is expressed in microglia [24]. As microglia activation is an important aspect of HFD-induced neuroinflammation, we decided to examine the changes in microglial marker transcripts in AKOs. The transcripts *Csf1r* and *Kcnk6* were both significantly upregulated (Table 4). *Csf1r* encodes for the receptor of colony stimulating factor 1, a positive regulator of the M2 microglial transcriptome [87]. KCNK6 is associated with a homeostatic microglial phenotype [98]. Other upregulated microglial transcripts include positive regulators of memory and long-term potentiation [92,93], and neuronal cell differentiation and survival [90,97]. Together HFD-fed AKO mice have changes in microglial transcripts consistent with homeostatic microglia and protection against cognitive decline. Future studies will include the immunohistochemical staining of these microglial markers to provide further insight on their regulation in the AKO hippocampus.

The HFD-fed FABP4 knockout model has revealed an altered hippocampal transcriptome that is shown to have changes in metabolic pathways and be neuroprotective against HFD-induced neuroinflammation and cognitive decline. We and others have shown that knockout of FABP4 leads to an alleviation of HFD-induced peripheral and central inflammation, insulin insensitivity, and cognitive decline [16,20,23]. In the present study, we have shown that this alleviation in the brain is associated with changes in metabolic pathways, Golgi transport, and WNT/β-Catenin signaling. As there is much supporting evidence that the WNT/β-Catenin pathway is neuroprotective against cognitive decline, our study suggests a role for WNT/β-Catenin in the FABP4 knockout-induced alleviation of cognitive decline. Other studies have shown that WNT/β-Catenin has a role in the regulation of microgliosis. A decrease in WNT/β-Catenin signaling has been shown to lead to proinflammatory microglial activation in the developing brain [125]. Wnt-3a, an activator of WNT/β-Catenin signaling, causes a decrease in the expression of inducible nitric oxide synthase (iNOS) and TNF-α and a decrease in microgliosis [126], while increasing insulin sensitivity [127]. Other activators of WNT/β-Catenin signaling have been shown to cause a switch in microglia from a proinflammatory phenotype to an anti-inflammatory one after ischemic stroke [128]. As many targets of WNT/β-Catenin were found to be unchanged in this study, further work is necessary to determine the mechanism and role of WNT/β-Catenin signaling in the alleviation of HFD-induced neuroinflammation and cognitive decline caused by FABP4 knockout. These studies will likely focus on isolated microglia from whole brain tissue. Single-cell RNA sequencing (scRNA-seq) has been used to define microglial immunoheterogeneity in models of AD [129,130]. Future studies using scRNA-seq to elucidate the microglial transcriptome changes that occur with FABP4 knockout in HFD-fed mice will provide further clarity on the mechanisms that lead to neuroprotection.

## 4. Materials and Methods

### 4.1. Mouse Model of Obesity and Immunometabolism

The FABP knockout mouse model (also known as aP2^−/−^ or AKO mouse) carries a null mutation in FABP4 bred to wild-type C57BL/6J mouse background [16]. We refer to this transgenic mouse line (and tissue derived from these mice) as AKO mice. Genotyping was confirmed using PCR as previously described [16,124]. Mice were maintained in a 12:12 h light/dark cycle in a temperature-controlled room (21–22 °C) and were group-housed. WT and AKO mice (*n* = 5 per genotype) were placed on a high fat diet (HFD; Research Diets D12492; 60% total fat and 32% saturated fat) for 12 weeks starting at 15 weeks of age and water was provided ad libitum. The experimental protocol was approved by the Institutional Animal Care and Use Committee at the Minneapolis VAHCS.

### 4.2. Brain Dissection and RNA Isolation

Hippocampal tissue was rapidly dissected from WT and AKO mice sacrificed during the light phase, 5–8 h after lights on [131]. Total RNA was extracted from hippocampal tissue with the aid of Trizol (Invitrogen; Carlsbad, CA, USA), purified using the RNeasy Mini Kit (Qiagen, Hilden, Germany) and both procedures were performed according to the manufacturer’s specification [131,132].

### 4.3. RNA-Seq cDNA Library Synthesis

Total RNA (3 µg) samples were sent to Novogene (Davis, CA, USA) for RNA-seq analysis. Briefly, RNA purity (OD260/OD280) was quantified using a Nanodrop spectrophotometer (Thermo Scientific; Waltham, MA, USA), RNA integrity and potential contamination was analyzed using agarose gel electrophoresis, and RNA integrity was further analyzed using the Agilent 2100 bioanalyzer (Agilent Technologies, Santa Clara, CA, USA). mRNA was purified from total RNA using poly-T oligo-attached magnetic beads and mRNA was then fragmented randomly. All sequencing was performed in one batch. First strand cDNA was synthesized using random hexamer primer and M-MuLV Reverse Transcriptase Minus (RNase H−). Second strand cDNA synthesis was then performed using DNA Polymerase I and RNase H with dTTP replaced by dUTP. Double-stranded cDNA was purified using AMPure XP beads. Remaining overhangs of the purified double-stranded cDNA were converted into blunt ends via exonuclease/polymerase activities. After adenylation of 3′ ends of DNA fragments, NEBNext Adaptor with hairpin loop structure was ligated to prepare for hybridization. The second strand cDNA was then digested by USER enzyme. The final library was obtained by PCR amplification and purification of PCR products by AMPure XP beads. Library quality was ensured using the Agilent 2100 bioanalyzer (Agilent Technologies, Santa Clara, CA, USA). Clustering was performed on a cBot cluster generation system using the TruSeq PE Cluster Kit v3-cBot-HS (Illumina; San Diego, CA, USA). Library sequencing was performed using the Illumina HiSeq Platform (Illumina; San Diego, CA, USA).

### 4.4. RNA-Seq Analysis

Raw data (raw reads) were stored in FASTQ (fq) format. Raw reads containing adaptors, over 10% indeterminate bases, or over 50% low-quality bases were removed. Clean reads were mapped to a reference genome using STAR software. Differential expression analysis was performed using the DESeq2 R package. P-values were adjusted using Benjamini and Hochberg’s procedure. Enrichment analysis of differentially expressed genes was performed using clusterProfiler software (version 3.16), including GO, DO, KEGG and Reactome database enrichment [133].

### 4.5. Reactome Analysis

Transcript quantification was performed with Salmon 1.0 [134] packaged in clusterProfiler 3.14.2 [135]. It was run with mapping mode selective alignment, minimum count of three, and relative abundance calculated in transcripts per million (TPM) relative to reference RNA transcripts from GENCODE VM23. Overrepresentation of transcripts and pathways and all other expression analysis was conducted with Reactome gene set analysis (ReactomeGSA) with Reactome pathway browser 3.7, database release 74 using the gene set enrichment analysis (GSEA) method and including disease pathways [136]. Within ReactomeGSA, normalization was performed with edgeR’s calcNormFactors function, transformed using limma’s voom and normalizeBetweenArrays functions, and pathway analysis using limma’s camera function as implemented in the respective Bioconductor R package. Reactome identifiers are shown in square brackets and can be accessed at https://reactome.org. Unless otherwise noted, statistical significance was evaluated with a cutoff of ≤ 0.01 using the false discovery rate (FDR) as calculated using the Benjamini-Hochberg procedure for pathways and multiple correction adjusted *p*-values for transcripts (or genes). Transcript identifiers were translated to Ensembl gene IDs (names) with g:Profiler [137]. Additional ad hoc pathway analysis was conducted with GO, KEGG databases. Estimate surrogate cell type proportions variables were calculated with BRETIGEA [138]; input gene-level expression calculated by summing transcript TPMs per gene, and output with mouse species and separate runs with 20, 50, and 200 markers for all six cell types (astrocytes, endothelial cells, microglia, neurons, oligodendrocytes, and oligodendrocyte precursor cells). Senescence was evaluated with a list of 88 gene markers from the Tabula Muris Consortium [139].

### 4.6. Western Blotting

Western blotting was performed using dissected hippocampal tissues from AKO and WT mice as previously described. Tissue was homogenized in RIPA buffer (Thermo Scientific, Waltham, MA, USA) and PIC (HALT; Thermo Scientific, Waltham, MA, USA) using a tissue homogenizer (Bullet Blender; Next Advance, Troy, NY, USA). Nuclear fractions were isolated using a nuclear extraction reagent (NE-PER; Thermo Scientific, Waltham, MA, USA). Protein concentration was determined using an infrared spectrometer (Direct Detect; Millipore, Burlington, MA, USA). 25 µg of protein was loaded onto a 4-20%, 30 µL-well precast gel (Bio-Rad, Hercules, CA, USA). Protein was transferred using the Trans-Blot Turbo system (Bio-Rad, Hercules, CA, USA). Total protein was stained using Licor Revert 700 stain (LI-COR, Lincoln, NE, USA) and imaged using the LI-COR Odyssey FC imager (LI-COR, Lincoln, NE, USA). Blocking was performed using Intercept Blocking Buffer (LI-COR, Lincoln, NE, USA). Primary antibody was used at 1:2000 dilution for a concentration of 1.25 × 10^−1^ µg/mL (BD Biosciences 610153; BD Biosciences, Franklin Lakes, NJ, USA), performed overnight at 4 °C with agitation. Secondary antibody was used at 1:10,000 dilution for a concentration of 1.0 × 10^−1^ µg/mL (LI-COR 926-32210; LI-COR, Lincoln, NE, USA) and imaged. Densitometry was performed using Image Studio software (LI-COR, Lincoln, NE, USA) and statistical analysis was performed using *t*-test (GraphPad Prism 7; GraphPad, San Diego, CA, USA).

## Figures and Tables

**Figure 1 ijms-24-03381-f001:**
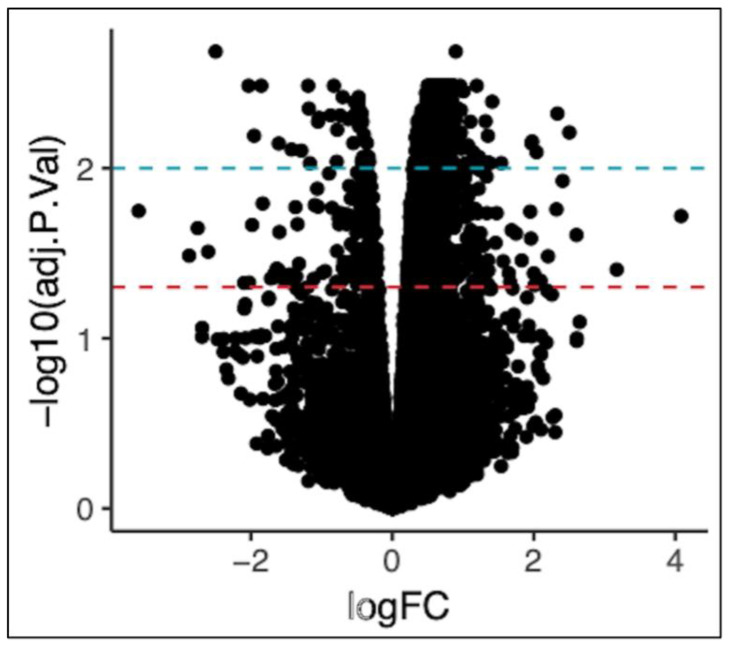
Volcano plot of the distribution of magnitude and significance of all genes. Log fold change (log2) is represented on the *x*-axis. Significance (log10 adjusted *p*-value) is represented on the *y*-axis. The red line indicates *p*adj = 0.05 and the blue line indicates *p*adj = 0.01.

**Figure 2 ijms-24-03381-f002:**
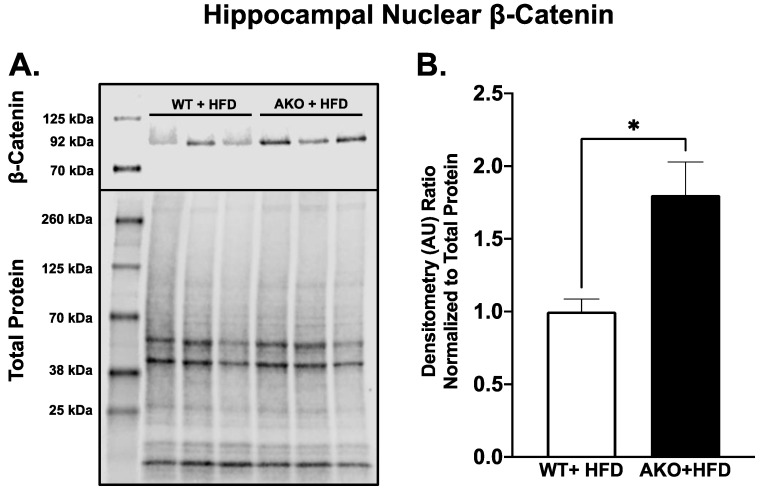
(**A**) Western blot of nuclear β-Catenin in HFD-fed AKO and WT mice. Licor Revert 700 total protein stain is shown below. (**B**) Densitometry normalized to total protein. Unpaired *t*-test, *n* = 3, * denotes significance of *p* < 0.05 (*p* = 0.03).

**Table 1 ijms-24-03381-t001:** Downregulated Transcripts in HFD-fed AKO Compared to HFD-fed WT Mice. Transcripts with LFC ≤ −1.00 and *p*adj ≤ 0.001 are displayed. Gene symbol, Ensembl ID, gene description, log fold change (log2), log10 adjusted *p*-value, and relevant known function are displayed.

Gene Symbol	Ensembl ID	Gene Description	Log FC	*p*adj	Function
*Fabp4*	ENSMUST00000029041	Fatty acid binding protein 4, adipocyte	−2.50	2.06 × 10^−3^	Positive regulator of obesity linked inflammation and ER stress [20]
*Sox6*	ENSMUST00000072804	SRY (sex determining region Y)-box 6	−2.03	3.27 × 10^−3^	Negative regulator of Wnt/β-catenin signaling [25]
*Apoa2*	ENSMUST00000005824	Apolipoprotein A-II	−1.95	6.44 × 10^−3^	Component of HDL with role in lipid metabolism and positive regulator of amyloidosis [32], Positively associated with cognitive decline [27]
*Ube3a*	ENSMUST00000202945	Ubiquitin protein ligase E3A	−1.85	3.27 × 10^−3^	Positive regulator of amyloid plaque formation [30], Positive regulator of GABA release [33]
*Tpx2*	ENSMUST00000028969	TPX2-microtubule associated	−1.61	7.15 × 10^−3^	Positive regulator of proinflammatory cytokines [28]
*Dmd*	ENSMUST00000114000	Dystrophin, muscular dystrophy	−1.43	7.76 × 10^−3^	Positive regulator of proinflammatory cytokines [29], Associated with altered GABA signaling [35]
*Eif4e2*	ENSMUST00000113233	Eukaryotic translation initiation factor 4E member 2	−1.29	7.88 × 10^−3^	Phosphorylated state is positively associated with hyperphosphorylated tau and AD [36]
*Zfp971*	ENSMUST00000108926	Zinc finger protein 971	−1.19	3.27 × 10^−3^	Unknown function, predicted to increase DNA-binding transcription repressor activity [37], contains KRAB box and C2H2 type domain [38], may have a role in neurodegenerative diseases [39]
*Zwint*	ENSMUST00000105431	ZW10 interactor	−1.18	4.46 × 10^−3^	Associated protein has role in retrograde trafficking from Golgi to ER [34]
*Tmod2*	ENSMUST00000164100	Tropomodulin 2	−1.17	9.35 × 10^−3^	Negative regulator of neurite outgrowth [26]
*Lonp2*	ENSMUST00000155433	lon peptidase 2, peroxisomal	−1.05	5.32 × 10^−3^	Upregulated in response to ER stress [31]
*Wtap*	ENSMUST00000159986	Wilms tumour 1 -associating protein	−1.04	5.05 × 10^−3^	Positive regulator of m6A methylation [40] promoting AD development [41]

**Table 2 ijms-24-03381-t002:** Upregulated Transcripts in HFD-fed AKO Compared to HFD-fed WT Mice. Transcripts with LFC ≥ 1.00 and *p*adj ≤ 0.001 are displayed. Gene symbol, Ensembl ID, gene description, log fold change (log2), log10 adjusted *p*-value, and relevant known function are displayed.

Gene Symbol	Ensembl ID	Gene Description	Log FC	*p*adj	Function
*Sort1*	ENSMUST00000135636	Sortilin 1	1.007	3.53 × 10^−3^	Positive regulator of murine apoE clearance in brain [55], Regulator of lipid metabolism in brain [56]
*Hspa5*	ENSMUST00000100171	Heat shock protein 5	1.038	9.34 × 10^−3^	Antiapoptotic regulator of ER stress and unfolded protein response [42]
*Adra2a*	ENSMUST00000237285	Adrenergic receptor, alpha 2a	1.069	7.32 × 10^−3^	Positive regulator of NMDA receptor-dependent long-term potentiation [46]
*Atp2a2*	ENSMUST00000031423	ATPase, Ca++ transporting, cardiac muscle, slow twitch 2	1.103	5.32 × 10^−3^	Regulator of calcium homeostasis [57], Downregulated in AD brain [58]
*S1pr3*	ENSMUST00000087978	Sphingosine-1-phosphate receptor 3	1.144	8.07 × 10^−3^	Negative regulator of inflammation in brain [45], Positive regulator of spatial working memory [49]
*Mdga1*	ENSMUST00000171691	MAM domain containing glycosylphosphatidylinositol anchor 1	1.151	8.87 × 10^−3^	Positive regulator of hippocampal long-term potentiation [47], Regulator of trans-synaptic bridge formation [59]
*H3c15*	ENSMUST00000167403	H3 clustered histone 15	1.193	3.27 × 10^−3^	Associated with the formation of the β-catenin:TCF transactivating complex and Senescence-Associated Secretory Phenotype (SASP) [60]
*Paqr3*	ENSMUST00000069453	Progestin and adipoQ receptor family member III	1.226	8.60 × 10^−3^	Regulator of cholesterol homeostasis [61], Regulator of autophagy [62]
*Brd1*	ENSMUST00000109380	Bromodomain containing 1	1.317	5.30 × 10^−3^	Positive regulator of H3K14 acetylation [63] associated with synaptic plasticity [50]
*Gpr68*	ENSMUST00000110066	G protein-coupled receptor 68	1.351	6.44 × 10^−3^	Positive regulator of adult hippocampal neurogenesis [52]
*Sufu*	ENSMUST00000111867	SUFU negative regulator of hedgehog signaling	1.355	9.26 × 10^−3^	Positive regulator of adult hippocampal neurogenesis [53]
*Grm1*	ENSMUST00000105560	Glutamate receptor, metabotropic 1	1.378	9.34 × 10^−3^	Positive regulator of pyramidal neuron excitation [64]
*Nsd3*	ENSMUST00000146919	Nuclear receptor binding SET domain protein 3	1.412	4.06 × 10^−3^	Positive regulator of REST mediated H3K36 trimethylation and associated with antiapoptotic genes [65]
*Dlgap1*	ENSMUST00000155016	DLG associated protein 1	1.538	9.33 × 10^−3^	Positive regulator of synaptic scaling in excitatory synapses [51]
*Grik2*	ENSMUST00000105484	Glutamate receptor, ionotropic, kainate 2 (beta 2)	1.969	7.15 × 10^−3^	Positive regulator of NMDA receptor-independent and KA receptor-dependent hippocampal long-term potentiation [48]
*Fbxw7*	ENSMUST00000107678	F-box and WD-40 domain protein 7	1.976	6.95 × 10^−3^	Negatively associated with glutamate mediated excitotoxicity and negative regulator of pro-apoptotic protein c-Jun [66]
*Nox1*	ENSMUST00000033610	NADPH oxidase 1	2.038	8.04 × 10^−3^	Positive regulator of Wnt/β-catenin signaling [54], Positive regulator of M2-type macrophage polarization [67]
*Smpd4*	ENSMUST00000090159	Sphingomyelin phosphodiesterase 4	2.334	4.77 × 10^−3^	Negatively associated with ER stress and apoptosis in brain [43]
*Zfp329*	ENSMUST00000121215	Zinc finger protein 329	2.502	6.16 × 10^−3^	Ortholog is a positive regulator of antiapoptotic protein BCL2 [44]

**Table 3 ijms-24-03381-t003:** Differentially Expressed Pathways in HFD-fed AKO Compared to HFD-fed WT Mice. FDR ≤ 0.01. Pathway name, stable identifier, log fold change (log2), and false discovery rate for each pathway are displayed. Within each pathway, gene ID, Ensembl ID, gene description, log fold change (log2), log10 adjusted *p*-value, and relevant known function are displayed.

Reactome Pathway	Gene Symbol	Ensembl ID	Gene Description	Log FC	*p*adj	Known Function
The citric acid (TCA) cycle and respiratory electron transport (R-HSA-1428517) LFC: 0.03 FDR: 2.71 × 10^−3^	*Ndufb2* *Ndufb4* *Ndufb11* *Ndufs8* *Ndufa4* *Ndufb9* *Ndufa2*	ENSMUST00000119379 ENSMUST00000023514 ENSMUST00000116621 ENSMUST00000237341 ENSMUST00000204978 ENSMUST00000022980 ENSMUST00000014438	NADH:ubiquinone oxidoreductase subunit B2 NADH:ubiquinone oxidoreductase subunit B4 NADH:ubiquinone oxidoreductase subunit B11 NADH:ubiquinone oxidoreductase core subunit S8 Ndufa4, mitochondrial complex associated NADH:ubiquinone oxidoreductase subunit B9 NADH:ubiquinone oxidoreductase subunit A2	−0.32 −0.29 −0.28 −0.27 −0.25 −0.23 −0.20	2.39 × 10^−2^ 2.89 × 10^−2^ 3.32 × 10^−2^ 4.63 × 10^−2^ 4.69 × 10^−2^ 4.16 × 10^−2^ 4.83 × 10^−2^	Partial inhibition of Complex I leads to reduced oxidative stress and increased long-term potentiation in AD mice [68]
	*Cox7b*	ENSMUST00000033582	Cytochrome c oxidase subunit 7B	−0.31	1.13 × 10^−2^	Loss leads to decrease in oxidative stress and amyloid formation in AD mice [71]
	*Uqcrh*	ENSMUST00000078676	Ubiquinol-cytochrome c reductase hinge protein	−0.29	3.04 × 10^−2^	Positive regulator of apoptosis via cytochrome c release [72]
	*Atp5j2* *Atp5h*	ENSMUST00000161741 ENSMUST00000043931	ATP synthase, H+ transporting, mitochondrial F0 complex, subunit F2 ATP synthase, H+ transporting, mitochondrial F0 complex, subunit D	−0.27 −0.26	1.82 × 10^−2^ 3.98 × 10^−2^	Inhibition is neuroprotective in aging and AD [86]
	*mt-Nd3*	ENSMUST00000082411	Mitochondrially encoded NADH dehydrogenase 3	−0.23	4.06 × 10^−2^	Partial inhibition of Complex I leads to reduced oxidative stress and increased long-term potentiation in AD mice [68]
	*mt-Co3*	ENSMUST00000082409	Mitochondrially encoded cytochrome c oxidase III	−0.21	3.87 × 10^−2^	Loss leads to decrease in oxidative stress and amyloid formation in AD mice [71]
	*mt-Cytb*	ENSMUST00000082421	Mitochondrially encoded cytochrome b	0.21	4.53 × 10^−2^	Decreased in AD brain [73]
	*Acad9*	ENSMUST00000011492	Acyl-Coenzyme A dehydrogenase family, member 9	0.29	2.59 × 10^−2^	Role in fatty acid oxidation, downregulation associated with neurological disease [69]
	*Aco2*	ENSMUST00000023116	Aconitase 2, mitochondrial	0.29	1.42 × 10^−2^	Suppressed by nitric oxide (Palmieri et al., 2020), Decreased in AD patient lymphocytes [76]
	*Sdha*	ENSMUST00000022062	Succinate dehydrogenase complex, subunit A, flavoprotein (Fp)	0.29	1.02 × 10^−2^	Downregulation is associated with oxidative stress and insulin resistance [74]
	*Ogdh*	ENSMUST00000003461	Oxoglutarate (alpha-ketoglutarate) dehydrogenase (lipoamide)	0.30	3.73 × 10^−2^	Inhibited by lipopolysaccharide and IFN-γ stimulation in macrophages [77]
	*Cs*	ENSMUST00000005826	Citrate synthase	0.32	1.53 × 10^−2^	Downregulated in AD patient platelets [78]
	*Pdk2*	ENSMUST00000038431	Pyruvate dehydrogenase kinase, isoenzyme 2	0.35	4.52 × 10^−2^	mRNA levels modulated during aging in brain [79]
	*Timmdc1*	ENSMUST00000002925	Translocase of inner mitochondrial membrane domain containing 1	0.36	9.17 × 10^−3^	Loss is associated with axonal neuropathy and cognitive decline [70]
	*Taco1*	ENSMUST00000002048	Translational activator of mitochondrially encoded cytochrome c oxidase I	0.40	3.68 × 10^−2^	Loss associated with motor disfunction and mitochondrial disease in mice [75]
Disassembly of the destruction complex and recruitment of AXIN to the membrane (R-HSA-4641262) LFC: 0.38 FDR: 7.21 × 10^−3^	*Ppp2cb* *Ppp2r5a* *Ppp2r5e* *Ppp2r1a*	ENSMUST00000009774 ENSMUST00000067976 ENSMUST00000021447 ENSMUST00000007708	Protein phosphatase 2 (formerly 2A), catalytic subunit, beta isoform Protein phosphatase 2, regulatory subunit B′, alpha Protein phosphatase 2, regulatory subunit B′, epsilon Protein phosphatase 2, regulatory subunit A, alpha	0.25 0.26 0.31 0.37	1.94 × 10^−2^ 4.25 × 10^−2^ 2.85 × 10^−2^ 3.87 × 10^−2^	Positive regulator of Wnt/β-catenin signaling [80], Inhibition leads to spatial memory impairment [81]
	*Csnk1a1* *Csnk1g2*	ENSMUST00000165123 ENSMUST00000085435	Casein kinase 1, alpha 1 Casein kinase 1, gamma 2	0.26 0.33	3.90 × 10^−2^ 4.73 × 10^−2^	Negative regulators of SMAD3 and TGF-β signaling [82]
	*Ctnnb1*	ENSMUST00000007130	Catenin (cadherin associated protein), beta 1	0.37	1.38 × 10^−2^	Positive regulator of Wnt/β-catenin signaling, neuronal survival and synaptic plasticity, negative regulator of Aβ production [83]
	*Gsk3b*	ENSMUST00000023507	Glycogen synthase kinase 3 beta	0.52	3.62 × 10^−3^	Regulator of Wnt/β-catenin signaling, loss leads to synaptic and social defects in mice [84]
	*Dvl3*	ENSMUST00000003318	Dishevelled segment polarity protein 3	0.72	2.23 × 10^−2^	Positive regulator of Wnt/β-catenin signaling, downregulated in AD brain [83]
	*Fzd1*	ENSMUST00000054294	Frizzled class receptor 1	0.73	3.81 × 10^−3^	Positive regulator of Wnt/β-catenin signaling, loss leads to impairment of neuronal differentiation [85]

**Table 4 ijms-24-03381-t004:** Differentially Expressed Microglial Markers in HFD-fed AKO Compared to HFD-fed WT Mice. Transcripts with *p*adj ≤ 0.05 are displayed. Gene symbol, Ensembl ID, gene description, log fold change (log2), log10 adjusted *p*-value, and relevant known function are displayed.

Gene Symbol	Ensembl ID	Gene Description	Log FC	*p*adj	Function
*Hpgd*	ENSMUST00000034026	Hydroxyprostaglandin dehydrogenase 15 (NAD)	−0.337	3.60 × 10^−2^	Increased in aged tissues and negative regulator of PGE2 signaling [94]
*Cx3cr1*	ENSMUST00000064165	Chemokine (C-X3-C motif) receptor 1	0.393	1.08 × 10^−2^	Decreased in hippocampal tissue of diet-induced obese mice [95], Positive regulator of hippocampal long-term potentiation [92]
*Cd68*	ENSMUST00000018918	CD68 antigen	0.441	2.42 × 10^−2^	Marker of M1 and M2 microglial activation [96]
*Csf1r*	ENSMUST00000025523	Colony stimulating factor 1 receptor	0.530	9.83 × 10^−3^	Receptor of positive regulator of M2 microglia transcriptome [87]
*Cyth4*	ENSMUST00000043069	Cytohesin 4	0.571	1.90 × 10^−2^	Positive regulator of ARF1 [88] which is a regulator of ER/Golgi transport [89]
*Ccr5*	ENSMUST00000111442	Chemokine (C-C motif) receptor 5	0.584	9.10 × 10^−3^	Positive regulator of neuronal cell differentiation [97], Positive regulator of dopaminergic neuronal survival [90]
*Pla2g15*	ENSMUST00000034377	Phospholipase A2, group XV	0.611	3.81 × 10^−3^	Positively associated with long-term memory and downregulated in AD brain [93]
*Kcnk6*	ENSMUST00000085818	Potassium inwardly rectifying channel, subfamily K, member 6	0.698	3.99 × 10^−2^	Positively associated with homeostatic microglia [98]
*Lcp2*	ENSMUST00000052413	Lymphocyte cytosolic protein 2	0.788	3.68 × 10^−2^	Positively associated with FcγR-dependent phagocytosis [91]
*Blnk*	ENSMUST00000054769	B cell linker	1.567	3.52 × 10^−2^	Positively associated with FcγR-dependent phagocytosis [91]

## Data Availability

Datasets generated during this study are the property of the U.S. Department of Veterans Affairs and will be made available upon request.

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
