# Peer review of "RNAseq Analysis of FABP4 Knockout Mouse Hippocampal Transcriptome Suggests a Role for WNT/β-Catenin in Preventing Obesity-Induced Cognitive Impairment"

_ijms, 2023, doi:10.3390/ijms24043381_

Round 1

Reviewer 1 Report

The manuscript “RNAseq Analysis of FABP4 Knockout Mouse Hippocampal Transcriptome Reveals a Role for WNT/β-Catenin in Preventing Obesity-Induced Cognitive Impairment” shows RNA-seq analysis results using hippocampal tissues from FABP4 knockout and WT mice fed with high-fat diet (HFD). The authors demonstrated that HFD-fed FABP4 knockout mice had a hippocampal transcriptome, expressed in TPM relative to RNA transcripts, with lower RNA levels of FABP4, SOX6, APOA2, and others, and with higher RNA levels of ZFP329, SMPD4, NOX1 among others, using unknown method-based multiple correction p-values lower than 0.01. The authors used a p-value lower than 0.01 but no threshold for fold-change of RNA levels to select gene sets for pathways analysis, using ReactomeGSA with the Benjamini-Hochberg false discovery rate (FDR) lower than 0.01. The data showed that the pathways with the highest overall fold-change included “Disassembly of the destruction complex and recruitment of AXIN to the membrane,” “Processing of intronless Pre-mRNAs,” and “Transport to the Golgi and subsequent modification.” The authors also demonstrated that HFD-fed FABP4 knockout mice had hippocampal tissue with up-regulated RNA levels of BLNK and LCP2, among others, that are microglial markers (adjusted p-value < 0.05). Finally, the authors performed a western blot experiment to test the beta-catenin expression in the hippocampal nucleus in HFD-fed WT and in HFD-fed FABP4 knockout.

Here are a few comments that may clarify the manuscript.

Major comments:

1. In the list of DEG that the authors argue/imply to be positive regulators of Wnt/beta-catenin signaling (Table 3 or line 244), casein kinase 1 and GSK3B are arguably negative regulators for beta-catenin-based signaling as they promote proteolysis of beta-catenin in the canonical pathway. Having both positive and negative regulators of the signaling pathway in FABP4 knockout tissues is puzzling to interpret. The authors should justify how they interpret them as a positive regulator of the Wnt/beta-catenin signaling pathway.

2. The authors argue that Wnt/beta-catenin signaling has a role in reducing neuroinflammatory phenotype in HFD-fed FABP4 mutant mice. Wnt/beta-catenin signaling has a broad list of target genes for transcriptional regulation, but FABP4 knockout tissues did not show clear evidence of up-regulation of this target gene expression. Thus, the authors should justify why the tissue does not have Wnt/beta-catenin target genes in its DEG list.

Minor comments:

1. In pathway analysis, the authors did not use a threshold or at least not clearly state a threshold of RNA level fold changes for determining differentially expressed genes (DEG). Log2FC value 0.2 or 0.3 for DEG seems quite low for one to confidently argue that it is a meaningful difference. Hence the authors should justify why they chose not to mention Log2FC threshold or how they chose a low DEG threshold for this analysis.

Reviewer 2 Report

So et al. have prepared a manuscript utilizing RNAseq analysis of hippocampal tissue from FABP4 knockout mice fed a high fat diet for 12 weeks. This approach was based on their previous findings that obese FABP4 knockout mice exhibited decreased neuroinflammation and cognitive decline. The authors performed pathway analysis and differentially expressed genes. The authors claim that FABP4 reduction resulted in decreased neuroinflammation and cognitive decline induced by a high fat diet. The authors used an FABP4 knockout mouse model, which have decreased TNF-alpha, thus they hypothesized that it FABP4 plays a role in regulating high fat diet induced cognitive decline. This work utilized RNA-seq to identify differentially expressed genes and pathways in AOX knockout mice under a high fat diet, and have convincingly shown that neuroinflammatory, ER stress, etc pathways have decreased, along with an increase in B-catenin levels.

 Clarification

Page 2 - Their previous work showed that the FABP4 knockout mice have neuroprotective hippocampal transcriptomes that correlate with decrease in proinflammatory signaling, ER stress, apoptosis, etc this should be cited.

 Page 2- “AKO and WT mice were fed 60% HFD for 12 weeks 68 starting at 15 weeks of age”, authors should give rationale on why this timeframe used.

 Page 9 – The authors looked at microglial markers via their expressed transcripts, using western blotting, immunofluorescence to stain for total microglia and some of these markers would be impactful to show the change in transcription leads to a change in translation.

 Page 11 – Authors used Ponceu S total protein stain to normalize B-catenin bands with total cell protein, yet the western blot has been cropped. Full western blot for the total protein bands should be presented. Using an ELISA may also be more quantitative, as the western blot bands for the WT+HFD have a large variance in signal between the three samples they used.

 Page 11 – “146 The AKO model has shown an altered hippocampal transcriptome with changes in 147 metabolic pathways. AKOs had a decrease in pathways relating to the electron transport 148 chain (ETC)” – citation needed or clarification that they are referring to this study.

  Page 13 – Authors should consider adding a western blot or ELISA for FABP4 expression levels in order to strengthen their argument for both the HFD AKO and WT models.

 Page 14 – “We and others have shown that knockout of FABP4 leads to an alleviation of HFD-induced peripheral and central inflammation, insulin insensitivity, and cognitive decline.” – citation needed

 Page 14 – Authors state lipid metabolism pathways are altered in the knockout models; this should be expanded upon further in the discussion rather than leaving it as a list of genes and pathways in results.

Reviewer 3 Report

This is a well written and well designed study with results applicable to further research. The use of knockout animals is a useful tool for believable results. I cannot find any substantial areas to criticize. I do recommend that the authors take another look at the references-some titles are capitalized and others not.

Reviewer 4 Report

I would like to thank the Authors of the Manuscript "RNAseq Analysis of FABP4 Knockout Mouse Hippocampal Transcriptome Reveals a Role for WNT/β-Catenin in Preventing Obesity-Induced Cognitive Impairment" for their work and the Editors for the opportunity to provide commentary.

To my understanding, the Authors have performed a transcriptome analysis of the hippocampus for 5 WT mice and 5 KO mice for the FABP4 gene (AKO mice), the removal of which is known to decrease neuroinflammation and cognitive decline. Mice were given a high fat diet for 12 weeks from the age of 15 weeks. RNAseq, blotting and in silico transcriptome analysis reveal significantly downregulated transcripts in the AKO mice, associated with negative regulation of neurite outgrowth and WNT/catenin signaling, but positive regulation of cognitive decline, amyloid plaque formation and inflammation/ER stress, among others. There is also up regulation fo transcripts positively associated with synaptic plasticity, working memory and neurogenesis. Differentially expressed pathways related to WNT/catenin signaling were upregulated in AKO mice.

The manuscript reveals an interesting perspective in the relationship between FABP4 gene, a high-fat diet and hyppocampal transcriptome alteration in the KO mice, and what they have presented so far is very intriguing, but I am afraid that the Authors leave out several crucial points from the analysis.

Firstly, the Authors do not explore the relationship between AKO mice fed a high fat diet and those being fed a standard diet, in order to verify again that it is not simply the removal of a gene, but its interaction with diet, that defines the variation in hyppocampal transcriptome. Moreover, it would be interesting to explore the degree of transcriptome variation across multiple levels of "high fat" diets, but this is an observation beside the point of the paper.

Secondly, the Authors do not test experimentally the entity of cognitive decline, or if there actually is any, nor the change in metabolic conditions, but ground their discussion on the possibility that the differential expression of genes and metabolic pathways they observe is directly linked to the expected phenotype. This can be done at a theoretical and speculative level, but in order to affirm that the WNT/catenin signaling pathway has a role in preventing obesity-induced cognitive impairment, it has to be demonstrated in practice.

Overall, the quantity of work effectively done and of data presented are not enough to warrant a full article, as more experimental results are required in order to support the actual relationship the Authors are suggesting.

In an attempt to be fully transparent with the Authors of the Manuscript, I am going to write here in the comments what I would suggest to the Editors: either a major revision of the paper, if the Authors think they can add more content to support their claims (even though I understand that it cannot be done without re-doing everything, since the test mice from this experiment are gone) or rework the content in a less speculative way, or that it be classified as a Communication, but the title needs to be changed because the work did not actually prove what the Authors claim (it would be enough to change "reveal the role" with "suggest a possible involvement").

Round 2

Reviewer 4 Report

I would like to thank the Authors and Editors for the opportunity to comment on this revised version of the Manuscript.

The answers provided in regards to my previous comments are satisfactory, as are the changes in the main text, which is much stronger and easier to read after this round of revision. I appreciate that my point around the title being a bit too strong has been accepted and the very interesting discoveries are suggestive of the role of WNT/beta-catenin in preventing obesity-induced cognitive impairment.